# Orbitool: A software tool for analyzing online Orbitrap mass spectrometry data

Runlong Cai[1,*], Yihao Li[2,3,*], Yohann Clément[4], Dandan Li[5], Clément Dubois[5], Marlène Fabre[5], Laurence Besson[5],

Sebastien Perrier[5], Christian George[5], Mikael Ehn[1], Cheng Huang[2], Ping Yi[3], Yingge Ma[2], Matthieu Riva[5]

[1] Institute for Atmospheric and Earth System Research / Physics, Faculty of Science, University of Helsinki, Helsinki, 00140, Finland

[2] State Environmental Protection Key Laboratory of Formation and Prevention of Urban Air Pollution Complex, Shanghai Academy of Environmental Sciences, Shanghai, 200233, China

10 [3] School of Electronic, Information and Electrical Engineering, Shanghai Jiao Tong University, Shanghai, 200240, China

[4] Univ Lyon, Université Claude Bernard Lyon 1, CNRS, Institut des Sciences Analytiques, UMR 5280, 5 rue de la Doua, F-69100 Villeurbanne, France

[5] Univ. Lyon, Université Claude Bernard Lyon 1, CNRS, IRCELYON, F-69626, Villeurbanne, France

15 *: Contributed equally

*Correspondence to*: Matthieu Riva: matthieu.riva@ircelyon.univ-lyon1.fr and Yingge Ma: mayg@saes.sh.cn

## Abstract

The Orbitrap mass spectrometer has recently been proved to be a powerful instrument to accurately measure gas-phase and particle-phase organic compounds with a greater mass resolving power than other widely-used online mass spectrometers in atmospheric sciences. We develop an open-source software tool (Orbitool, https://orbitrap.catalyse.cnrs.fr) to facilitate the analysis of long-term online Orbitrap data. Orbitool can average long-term data while improving the mass accuracy by re-calibrating each mass spectrum, assign molecular formulae of compounds and their isotopes to measured signals, and export time series and mass defect plots. The noise reduction procedure in Orbitool can separate signal peaks from noise and reduce the computational and storage expenses. Chemical-ionization Orbitrap data from laboratory experiments on ozonolysis of monoterpenes and ambient measurements in urban Shanghai were used to test Orbitool. For the test dataset, the average mass accuracy was improved from < 2 ppm to < 0.5 ppm by mass calibrating each spectrum. The denoising procedure removed 97% of the noise peaks from a spectrum averaged for 30 min while maintaining the signal peaks, substantially helping the automatic assignment of unknown species. To illustrate the capabilities of Orbitool, we used the most challenging/complex dataset we have collected so far, which consists of ambient gas-phase measurements in urban Shanghai. These tests showed that Orbitool was able to automatically assign hundreds of molecular formulae as well as their isotopes with high accuracy.

# 1 Introduction

Biogenic and anthropogenic sources emit a wide variety of volatile organic compounds (VOCs) into the atmosphere (Hallquist et al., 2009; Shrivastava et al., 2017). Once emitted, VOCs can quickly react with different atmospheric oxidants (OH radicals, $O_3$, $NO_3$ radicals or Cl atoms) yielding a wide variety of oxidized VOCs (OVOCs) spanning a broad range of chemical formulae and, thus, volatilities (Hallquist et al., 2009; Li et al., 2020; Wennberg et al., 2018). OVOCs play a central role in the formation of atmospheric aerosols by either condensing onto pre-existing aerosol particles or by forming new particles (Hallquist et al., 2009; Jimenez et al., 2009; Kirkby et al., 2016; Shrivastava et al., 2017). Generally, the more oxidized OVOCs are, the lower their volatility, and the greater the probability to partition to the particle phase. However, the quantitative evaluation of the impact of aerosol on climate is yet inadequately constrained due to many factors, including an incomplete understanding of how VOC oxidation processes contribute to new particle and secondary organic aerosol formation (Glasius and Goldstein, 2016). Indeed, the gas phase oxidation of one single VOC can yield thousands of oxidized products (Glasius and Goldstein, 2016; Goldstein and Galbally, 2007; Li et al., 2020; Riva et al., 2019b). As a result, the chemical variety of OVOCs poses a major challenge in detecting, quantifying, and characterizing such a large number and wide variety of organic compounds.

Over the last decade, mass spectrometric techniques have made extensive improvements and are now well-suited to detect a large range of species simultaneously. This is highlighted by the key role of chemical ionization mass spectrometry (CIMS) in improving our understanding of the atmospheric chemical composition (Breitenlechner et al., 2017; Ehn et al., 2014; Jokinen et al., 2012; Krechmer et al., 2018; Lindinger et al., 1998; Yuan et al., 2017). Chemical ionization is a soft ionization technique where the analytes are ionized through a clustering process with the reagent ions and undergoes only minimal fragmentation. While CIMS provides very good sensitivities (i.e., as low as $10^4$ molecules $cm^{-3}$) (Jokinen et al., 2012) and is suitable to measure a wide variety of gaseous organic and inorganic compounds, they are mainly associated with time-of-flight (TOF) mass analyzers.

The mass resolving power of a TOF analyzer typically ranges from hundreds to less than 50 000 and the mass resolving power of online TOF-MS used in atmospheric measurements is only up to 15 000 (Riva et al., 2019b). Although often able to distinguish some isobaric species, these mass resolving powers still limit accurate assignment and quantification of OVOCs in a complex air sample. The existence of multiple overlapping ions

yields significant uncertainties (Cubison and Jimenez, 2015; Riva et al., 2019b; Stark et al., 2015). Computational approaches, including ion deconvolution procedures, are required to partly resolve this limitation in order to extract the maximum possible information content (Cubison and Jimenez, 2015; Meija and Caruso, 2004; Zhang et al., 2019, 2020). To overcome these limitations, we have coupled a high-resolution mass spectrometer (Orbitrap, Eliuk and Makarov 2015) with a chemical ionization source (Riva et al., 2019a, 2020) and an extractive electrospray

ionization inlet (Lee et al., 2020) for online analysis. Similarly, Zuth et al. (2018) have combined an atmospheric pressure chemical ionization (APCI) interface to an Orbitrap and have shown that such a technology can achieve much greater mass resolving power at a similar detection limit for the real-time measurement of organic aerosols. As a result, chemical ionization, extractive electrospray ionization, and/or APCI-Orbitrap represent very promising new techniques for online characterization of the chemical compositions of gaseous and particle phases at a very

high mass resolving power.

The demand for a new software tool for analyzing Orbitrap data comes along with applying an Orbitrap in long-term continuous online measurements. The commercial software, Xcalibur[TM] that has been used in our initial studies (Lee et al., 2020; Riva et al., 2019a, 2020), provides an interface for basic data analysis e.g., reading single or averaged scans and exporting the time evolution of selected signal peaks. However, more complicated data

analysis is usually needed for investigating the complexity and chemical processes occurring within the atmosphere. Hence, some customized software tools e.g., RawQuant (Kovalchik et al., 2018), RawTool (Kovalchik et al., 2019), and DIMSpy (Weber and Zhou, 2020) are developed to meet different demands. In atmospheric sciences, the concentrations of some key chemical species (e.g., peroxy radicals) in a typical atmospheric environment are

extremely low (< 1 ppt). The measured spectra must be averaged over a long period to decrease the noise level so that these low signals can be unambiguously identified among noises. This step implies averaging spectra across files while maintaining mass accuracy. Although facilitated by the high mass resolution of the Orbitrap, a minor proportion of the detected signal peaks still inevitably overlap with each other. Therefore, peak fitting is needed to

separate the overlapped signal peaks. In addition, using a list (i.e., "peak list") of possible species obtained in similar atmospheric environments reduces the expense for data analysis. The aforementioned features are already realized in existing software tools for analyzing data acquired with CIMS or proton-transfer-reaction (PTR) ToF-MS, e.g., TofTools (Junninen et al., 2010), Tofware, and other software developed to analyze PTR-TOF-MS dataset (Holzinger, 2015; Müller et al., 2013). It would seem straightforward to convert the Orbitrap raw data into certain

formats and then conduct data analysis using these existing software tools. However, due to the high mass resolution of the Orbitrap compared to TOF mass analyzers, the computational expense will be high if analyzing Orbitrap data using, for example, TofTools. This is because the TOF mass spectra are stored on equally spaced grids (with respect to TOF), with data points on the order of $10^4$-$10^5$. For comparison, approximately $10^8$ data points are needed to characterize a single Orbitrap spectrum using this grid-based data structure. Instead, saving only the

signals with their adjacent zeros will reduce the computational and storing expenses by orders of magnitude and the Orbitrap raw data is saved using this data structure. In addition, TofTools determines the noise level using the equally spaced data within a certain mass defect range, whereas such information is not recorded in the Orbitrap data. As a result, a new software tool for analyzing Orbitrap data is needed to facilitate its application in atmospheric researches.

In this study, we develop a new software tool, named Orbitool, for analyzing online Orbitrap mass spectrometry data. The working principles and features of Orbitool are introduced. Several test examples using both laboratory and atmospheric data are presented to illustrate and validate its features.

## 2 Orbitool

Orbitool is a software tool with a graphical user interface (GUI) for analyzing online Orbitrap mass spectrometry data. It is optimized for long-term field or laboratory campaigns and laboratory studies. Flexible input and output interfaces facilitate the coupling between Orbitool and other software. Orbitool's basic working principle is shown in Fig. 1, which includes data reading and averaging, noise determination and reduction, peak shape determination, mass calibration, signal abundance determination via peak fitting, molecular formula assignment, and data output. The details of these features will be elaborated in the following sections. Orbitool is mainly coded in Python with supports from several third-party libraries (Pedregosa et al., 2011; Goloborodko et al., 2013; Levitsky et al., 2018). To enhance computational speed, the mass spectra are processed using the Numba package (Lam et al., 2015) and chemical composition and signal abundance are determined using the Cython package (Behnel et al., 2011). RawFileReader from Thermo Fisher Scientific is used to read the raw data reported by Orbitrap. Orbitool was tested in Python 3.6 on the Microsoft Windows operating system. An executable version of Orbitool and its open-source codes are publicly available (see *Software availability*).

**2.1 Data averaging.** The data collected by the Orbitrap is stored in a RAW file, containing segmented peak intensities and their corresponding masses of each scan, scan filters, event logs, instrumental information, etc. The RAW file records only the spectra of all the detected peaks rather than a dense grid-based spectrum. Most tuning parameters of Orbitrap, e.g., injection time of each micro scan and the number of micro scans for every single scan, do not affect the analysis with Orbitool. However, it should be specially clarified that Orbitool is compatible with the profile mode but not the centroid mode of Orbitrap, because attributing chemical formulae to measured signal peaks is based on fitting peaks to the profile data.

The first step to analyze raw mass spectrometry data with Orbitool is reading and averaging the spectra. Since the abundances of most detected species can be extremely low, especially under typical ambient concentrations,

averaging is needed for identifying these signal peaks against noise. The raw data can be imported into Orbitool as single files or using a file folder. The user can choose to average all the imported files or a subset of them. The average spectra are calculated based on either scan number or time. For the time-based data averaging, time intervals are determined by the beginning time, the ending time, and the duration of each time interval. Orbitool calculates the average spectrum for each time interval. The beginning and end times can be specified by the user, while they are by default obtained from the time range of the input data files. The data recorded in the same file is averaged using an averaging function in the Thermo RawFileReader library and the average is calculated with respect to scan number. Hence, it is assumed that the parameters determining the duration of every single scan (e.g., injection time and the number of micro scans) are kept constant during each time interval. When multiple data files are found in the same time interval for averaging, the average spectra of each file are calculated first and then averaged with respect to time. Peaks close to each other in the averaged spectrum are merged or identified as separated peaks according to the mass resolution of the Orbitrap. To facilitate quick data analysis, averaging is not conducted until either the user or Orbitool calls the averaged spectrum in order to minimize unnecessary expenses. Further, data averaging can be skipped. In this case, single spectra are used instead of averaged spectra in the following steps. Orbitool also supports filtering data using the ion polarity. For instance, when the positive polarity is selected, only spectra from the positive mode are taken for the averaging while spectra from the negative mode are excluded. Hence, this feature can support the analysis of Orbitrap measurements that switch the ion polarity routinely (e.g., every 1 minute) to realize near real-time measurement of both positively and negatively charged compounds.

**2.2 Noise determination and reduction.** The noise is estimated and then removed from the average spectra during this step. When converting electronic signals into a spectrum via the Fourier transform, the instrument removes signals below the noise threshold defined by the Orbitrap. Signals exceeding this noise threshold are converted into peaks recorded in the data file. Since this procedure does not remove all the noise signals, some noise peaks are

also recorded in the data file. In this study, "noise" refers to these noise peaks. It is difficult to separate the signal peaks for some compounds with low concentrations from noise peaks. However, after averaging for a sufficiently long period, the signal peaks usually exceed noise peaks because averaging reduces the height of noise peaks but not signal peaks. To estimate the noise level, Orbitool first takes all the detected peaks whose peak positions are located in the mass defect range of [0.5, 0.8] Da. Mass defect is the absolute difference between the accurate mass and the nominal mass of the given compound, which is determined by the type and quantity of atoms contained in a compound and its ion polarity. Such a mass defect range is chosen because most of the observed compounds in atmospheric measurements are located outside of this mass range (Zhang et al., 2019). Then, peaks lower than a certain percentile of the peak heights in this mass defect range (0.5-0.8) are taken as noise peaks. This certain percentile can be customized and it is by default 70%, i.e., the 70% low peaks within this mass defect range are taken as noise peaks and used to estimate the noise level. The noise and discriminator levels are determined as $\mu$ and $\mu + 3\sigma$, respectively; where $\mu$ and $\sigma$ are the mean and standard deviation of these noise peaks, respectively. The noise and discriminator levels as a function of mass can be exported into files for further analysis.

Concerning different amounts of noise peaks in an averaged spectrum, Orbitool provides three strategies to deal with the obtained noise: a) remove noise peaks below the discriminator level; b) remove noise peaks below the discriminator level and subtract noise level ($\mu$) from all signals above the discriminator level; c) preserve all the noise peaks. Option a) is the default setting. Option b) is for averaging over a long period so that the signal peaks are likely to be overlapped by noise peaks and hence the abundances of signal peaks have contributions by noise. Option c) is for averaging over a very short period so that it is difficult to distinguish signals from noise. In addition to correcting the impact of noise to signal abundance, removing noise peaks reduces the expenses for both peak fitting and data storage. This is because each averaged spectrum in Orbitool is stored as a series of peaks rather than the signal intensities at each given mass grid.

**2.3 Single peak shape determination.** Although the mass resolution of the Orbitrap (e.g., 140 000 at mass 200 Da) is considerably higher than those of conventional online mass spectrometers in atmospheric research (Riva et al., 2019), some peaks (e.g., $^{13}C$ or $^{18}O$ isotopes) in an Orbitrap spectrum might still need to be distinguished via fitting multiple peaks to the measured signals. A known single peak shape is necessary for the peak fitting, where a single peak herein refers to the signals contributed by compounds with a single molecular formula. A signal peak hereafter refers to a measured peak, which may be composed of a single ion or multiple different ions. Orbitool uses a normal distribution to characterize the single peak shape (See Fig. 2A). To obtain the standard deviation of the normal distribution at different masses, Orbitool first takes the x highest signal peaks in the spectrum, where x can be set from 1 to 999 and it is 100 by default. The mass resolution of Orbitrap is inversely proportional to $\sqrt{m}$ (Zubarev and Makarov, 2013) where $m$ is mass; hence, the peak width is theoretically proportional to $\sqrt{m}$. According to this relationship, the measured widths of these selected peaks are normalized to widths at 200 Da by dividing them by $\sqrt{m/200\ \mathbf{Da}}$. 200 Da is chosen as the reference value because the instrumental resolution of the Orbitrap is usually reported at this mass. After this normalization, the mass resolution at 200 Da is determined via the least square fitting. The shape of some selected signal peaks composed of multiple ions may deviate from others. The user can remove these peaks manually on the GUI before the fitting. As shown in Fig. 2A, the normalized peaks can be well characterized using a normal distribution and the fitted resolution at 200 Da is close to the instrumental mass resolution.

**2.4 Mass calibration.** Due to the high mass resolving power of the Orbitrap, determining the mass accurately is critical to assign the molecular formula corresponding to a measured peak. The Orbitrap is able to maintain its mass accuracy (i.e., < 2 ppm) for long-time series with the help of constantly existing peaks with relatively high abundances, named lock masses (Olsen et al., 2005). Despite this feature, additional mass calibrations might be

needed, especially for long-term measurements, to ensure an optimal assignment of molecular formulae across the entire mass range. The user can also skip the mass calibration.

During the mass calibration step, Orbitool can automatically find the peaks of user-specified calibration species within a given uncertainty range. The mass positions of these calibration species are shown on the top of the averaged spectrum on the GUI to guide the eyes. Then, a calibration curve is fitted using a polynomial function and shown on the GUI. The users can set the degree of the polynomial and adjust the calibration curve by using different calibration species. The spectra saved in the same raw data file share the same calibration parameters. The difference between the measured mass and the calibration mass inferred from the chemical formula is shown on the GUI. If the analyzed dataset contains more than one raw file, the trends of these differences for all the calibration species are displayed (Fig. 2B). The calibration information can also be exported for further analysis. Using lock mass(es) during Orbitrap measurements can improve mass accuracy but it requires a constantly existing compound in the measured spectra. When charging compounds by chemical ionization, the reagent ion and its dimer are usually in high abundance. Therefore, it is recommended to use the reagent ions as lock masses when using a CI-Orbitrap. Instead of excluding the lock masses from the spectra (i.e., the default setting in Xcalibur), Orbitool treats the lock masses as normal signals since the abundance of the reagent ion is usually a critical parameter to characterize the performance of a CI-mass spectrometer.

**2.5 Peak assignment.** Identifying the abundances and assigning molecular formulae of compounds contained in each signal peak are main advantages of Orbitool. Before peak assignment, the user can merge the peaks in a given mass range using a customized mass tolerance. A merge sort algorithm (Knuth, 1998) is used to improve computational efficiency. The first step of peak assignment is to fit single peaks to the calibrated averaged spectrum. Since the width of a certain single peak is determined by its position (mass), the fitted parameters are only peak position and peak area (which can be converted to peak height). Orbitool usually conducts a default peak

fitting to the whole spectra. The number of single peaks used to fit a signal peak is determined automatically according to the number of local maxima of intensities. After a default fitting, the user can look into each signal peak. The signal peak, fitted single peaks, and fitting residues are shown on the GUI to indicate the quality of the default fitting. Orbitool supports repeating the peak fitting for each signal peak with a user-determined single ion

number, and the updated fitting results for the signal peak will replace the default fitting results in the whole spectra.

The possible molecular formulae corresponding to each fitted single peak is retrieved during fitting. Prior to assigning the molecular formulae from a given mass, the user can customize the possible elements contained in the detected compounds and the ion polarity. The minimum and maximum numbers of atoms contained in each measured ion, the range of equivalent double bond number, and the maximum tolerance of mass precision are used

as boundary conditions. Orbitool searches for all possible chemical formulae satisfying these boundary conditions for a given mass. If needed, the nitrogen rule can be used to help constraining the peak identification (Pellegrin, 1983). An empty string is returned if no possible chemical formula is found under the given conditions. The user can edit the retrieved chemical formulae after this automatic screening. A separate window on the GUI also supports the conversion of chemical formulae to mass at any analyzing step and vice versa.

Orbitool also supports the identification of isotopes. The users can adjust the possible isotopes in an isotope list and all the isotopes in this list will be considered. Using the default algorithm for peak assignment, the molecular formula containing a less abundant isotope can be determined only when its corresponding formulae containing the most abundant isotope is found in the spectrum. In addition, this default algorithm also checks whether the abundance ratio of these two species is consistent with their natural abundances within the uncertainty range. To

facilitate studies such as isotope labeling experiments, Orbitool also provides an algorithm that does not restrict isotope abundances during peak assignment. However, it is important to mention that the Orbitrap provides non-linear responses when the concentrations of the analytes are very low (i.e., $< 1 \times 10^6$ molecules cm$^{-3}$, at a 10-minute

integration time). As a result, the calculated isotope abundance may be substantially smaller than its true value (Riva et al., 2020).

After assigning molecular formulae to all the single peaks, a mass list is generated from the fitted peak position and their corresponding molecular formulae. The user may add all the molecular formulae into the mass list or select only a proportion of them. The unidentified chemical species are shown with empty strings. The mass list can be exported and saved into a csv file. Alternatively, an existing mass list with a certain format can be imported into Orbitool. The new mass list imported from a file or retrieved from a new mass spectrum can either substitute the existing list or merge with it. Orbitool supports fitting the signal peaks using a mass list instead of fitting all the signal peaks, which reduces the user's workload when analyzing the chemical species in a known system. For instance, when measuring ambient air, the user can average data over a whole day to obtain a mass list. After this, the user may restart data analysis from the beginning and average the raw data with a higher temporal resolution (e.g., 30 min). Then, the user may fit the signals using the former mass list derived from the whole day instead of repeating the default fitting and a manual correction.

**2.6 Time series and mass defect.** The time series (temporal evolution of abundance) of given chemical species can be calculated based on the measured time-evolving spectra. The chemical species for the time series are usually chosen from the mass list. Due to the unavoidable random error in the measured mass even after mass calibration, Orbitool searches for these chosen peaks within a customized uncertainty range (e.g., 1 or 2 ppm). The time series of all the chosen species are displayed on the GUI and they can be exported into other file formats. The temporal resolution of these time series is configured in the data averaging step. To facilitate a quick analysis, Orbitool can also show the time series of a given compound instead of using a mass list. The input for specifying this compound is either its accurate mass (or its molecular formula which will be automatically converted into accurate mass) with a mass tolerance or a mass range. Using this feature, the user can also obtain the abundance of all the signals within

a wide mass range. Orbitool is also able to show the mass defect plot on the GUI for an intuitive and rapid characterization of an averaged spectrum. The size of data points is determined by their intensity, which can be tuned on the GUI. These data points are colored by the number of a chosen element (e.g., C, O, N) or the double bond equivalent. The peaks without assigned molecular formulae can either be omitted or displayed in grey.

## 3 Results and Discussion

As illustrated in section 2.2, Orbitool estimates the noise level and discriminator level based on the statistics of the measured spectra. Except for the mass defect range of [0.5, 0.8] Da, no prior information is used for determining the noise level. This indicates that one cannot unambiguously distinguish every signal peak from the noise, which may theoretically introduce uncertainties to retrieve molecular formulae and abundances of the measured compounds. However, such uncertainties are practically negligible after averaging the spectrum. To validate the algorithm used in Orbitool for estimating the noise level, Fig. 3 shows the total number of peaks in each single or averaged spectrum measured in Shanghai. The noise and discriminator levels were first determined using a certain percentile of peaks in the mass defect range of [0.5, 0.8] Da. Peaks below the determined discriminator level in the entire mass range were then taken as noise and removed from the spectrum. Considering that some species may not be detected in a single scan (i.e., present at very low concentration), the total number of peaks in the averaged spectrum should grow with the increasing averaging time when the period is considerably short, and then it should converge to a certain value when the averaging period is sufficiently long. This is consistent with the observed trend of the total number of peaks after noise reduction in Fig. 3, in which the total number of peaks after noise reduction with the 50[th] percentile increases slightly with an increasing averaging time. In contrast, the total number of peaks without noise reduction increases dramatically with the increasing averaging time, indicating that most of the peaks observed in the averaged spectrum are noise. In addition, using the 25[th] or 70[th] percentile does not lead to a significant difference in the total number of peaks after noise reduction (within 5-10%), which indicates that

the noise level is governed by the instrument rather than the customized percentile value. Accordingly, Orbitool uses the 70th percentile as the default value to estimate the noise and discriminator levels.

Combining data averaging and noise reduction, Orbitool can remove most of the noise while maintaining most of the signal peaks. Figure 4 shows the averaged spectra before and after noise reduction measured from the ozonolysis of monoterpenes in ambient indoor air conditions and ambient air measured in Shanghai. The noise levels were calculated using the 70th percentile of peaks in the mass defect range of [0.5, 0.8] Da. They can also be identified by eye in the mass range from 600 Da to 750 Da, where only a few signal peaks were detected and noise peaks are at approximately the same height. The noise level in Fig. 4B is lower than that in Fig. 4A because it was averaged for a longer period. 97% of the measured peaks in the Shanghai data set after the 30-min averaging were identified as noise and hence removed. Since no significant mass dependencies of noise and discriminator levels were observed, Orbitool uses size-independent noise and discriminator levels. It should be clarified that some signal peaks with low abundances may be mistaken as noise peaks during noise reduction. Even after averaging for a longer period, some detected compounds with extremely low abundances may still be below the discriminator level.

To illustrate the capabilities of Orbitool, measured ions depicted in Fig. 4B were characterized. A total of 1850 molecular formulae were identified. Figure 5 shows the corresponding mass defect plot of gaseous compounds measured in urban Shanghai, whose data were retrieved and exported by Orbitool. Organic compounds contributed to a major proportion of the measured compounds. The reagent ion (i.e., $NO_3^-$) is not considered when sorting out the different ions into family groups. Hence the chemical analysis reveals that due to its higher mass resolving power compared to other online mass spectrometers, the CI-Orbitrap identifies a large number of gas-phase signals with elemental composition categories including CHO, CHON, CHONS, CHOS, and others. In addition, inorganic sulfuric acid was observed, which were reported to be the major gaseous precursors for nucleation in Shanghai (Yao et al., 2018). Interestingly, the presence of highly oxygenated CHONS and CHOS species might indicate the existence of uncharted chemical processes.

## 4 Summary

We developed a software tool named Orbitool for analyzing long-term online Orbitrap mass spectrometry data in atmospheric researches. Orbitool is capable of averaging raw data across files while improving the mass accuracy, distinguishing signal peaks against noise peaks, assigning chemical compounds and their isotopes via peak fitting,

and exporting time series and mass defect plots. Compared to similar software tools for atmospheric science studies such as TofTools, Orbitool redesigned the storage method for the measured signals so that it reduces the computational and storage expenses. The data measured in a laboratory study on ozonolysis of monoterpenes and in the ambient air of Shanghai were used to test Orbitool. The mass axis uncertainty after calibration using Orbitool was < 0.5 ppm in this test, which is lower than the average uncertainty of 2 ppm calibrated during measurements.

The noise reduction was demonstrated to be able to separate signal peaks from noise and reduce the amount of saved data. For instance, the total number of peaks of a test spectrum averaged over 30 min was reduced by 97%. The mass defect plot retrieved using Orbitool indicates that most of the chemical species in urban Shanghai measured by a CI–Orbitrap were organic compounds and reveals the presence of gaseous compounds with unique compositions (i.e., CHONS; CHOS).

Orbitool is open source and freely available for download (see Software availability below). We hope that it will evolve in the future with added features, optimally as a community effort as Orbitrap mass spectrometers become increasingly utilized as tools for online studies within, and possibly outside, atmospheric sciences.

### *Software availability*

An executable version of Orbitool and its source codes can be found at https://orbitrap.catalyse.cnrs.fr. The users

can either run Main.py via Python or Orbitool.exe without Python. The required Python environment (optional) for Orbitool is described in detail on the website. Orbitool will be continuously updated in the future. It is recommended to contact the corresponding authors for needs on software development.

*Author contributions*

M.R. initialized this study; Y.L., R.C. and Y.C. wrote the software with the help of Y.P.; D.L., C.D., Y.M., C.H., C.G. and M.R. conducted the experiments; D.L., C.D., S.P., Y.M., M.E., and M.R. tested Orbitool; M.R. analyzed the data; M.F. and L.B. developed the website; R.C. and M.R. wrote the manuscript with the contributions from all co-authors.

*Competing interests*

The authors declare that they have no conflict of interest.

*Acknowledgment*

This work was supported by the European Research Council (ERC-StG MAARvEL, grant nr 852161). M.R. wishes to thank the French National program LEFE (Les Enveloppes Fluides et l'Environnement), for their financial support. Financial supports from the National Key R&D Program of China (project number 2016YFC0200104 and 2018YFC0213800), Academy of Finland (project number 332547), and European Research Council via CHAPAs (850614) are appreciated. We gratefully acknowledge Jianmin Chen and Hui Chen (Fudan University) for their support during the field campaign. Finally, we would like to thank Bin-Yu Kuang (Zhejiang University) for useful discussions.

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

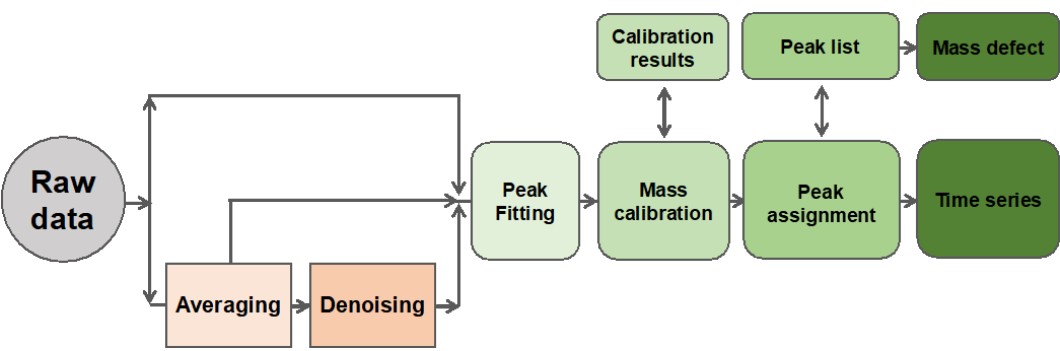

**Figure 1:** Simplified schematics of the working principle of Orbitool.

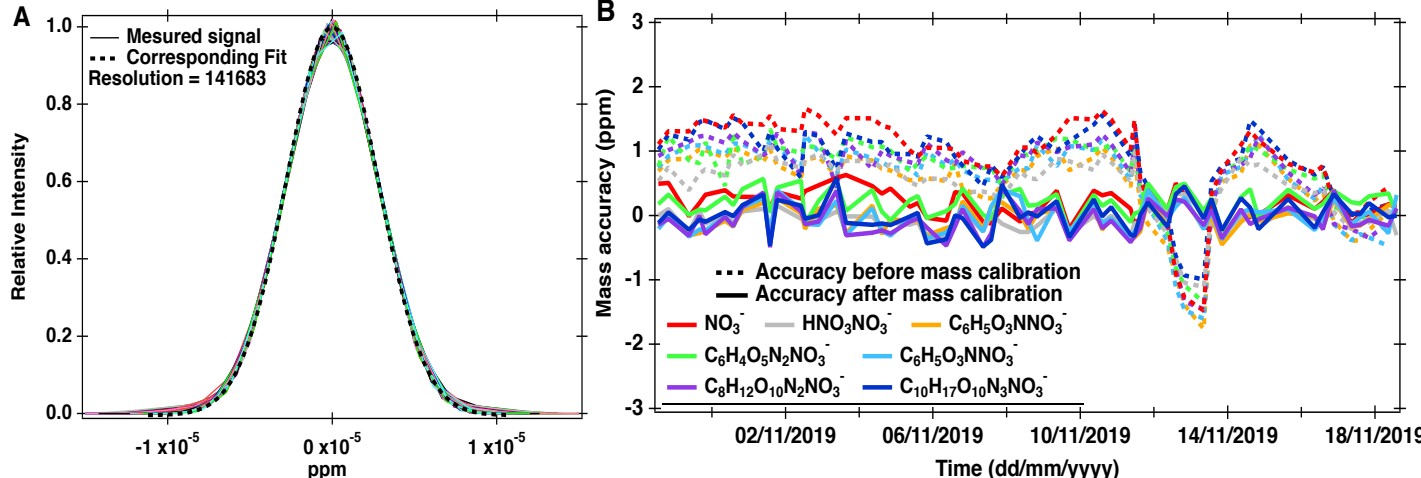

**Figure 2. (A)** Peak shape determined using the highest 100 peaks observed in the mass spectrometry. The position and width of signal peaks were normalized to 200 Da. The intensity of each signal peak was normalized by dividing by its peak intensity. **(B)** Mass accuracy of product ions at mass (in Da) 62 (red); 125 (grey); 183 (orange); 201 (green); 246 (blue); 358 (purple) and 401 (dark blue) before and after mass calibration. Each data point (as indicated by the fluctuation of the lines) corresponds to a raw data file.

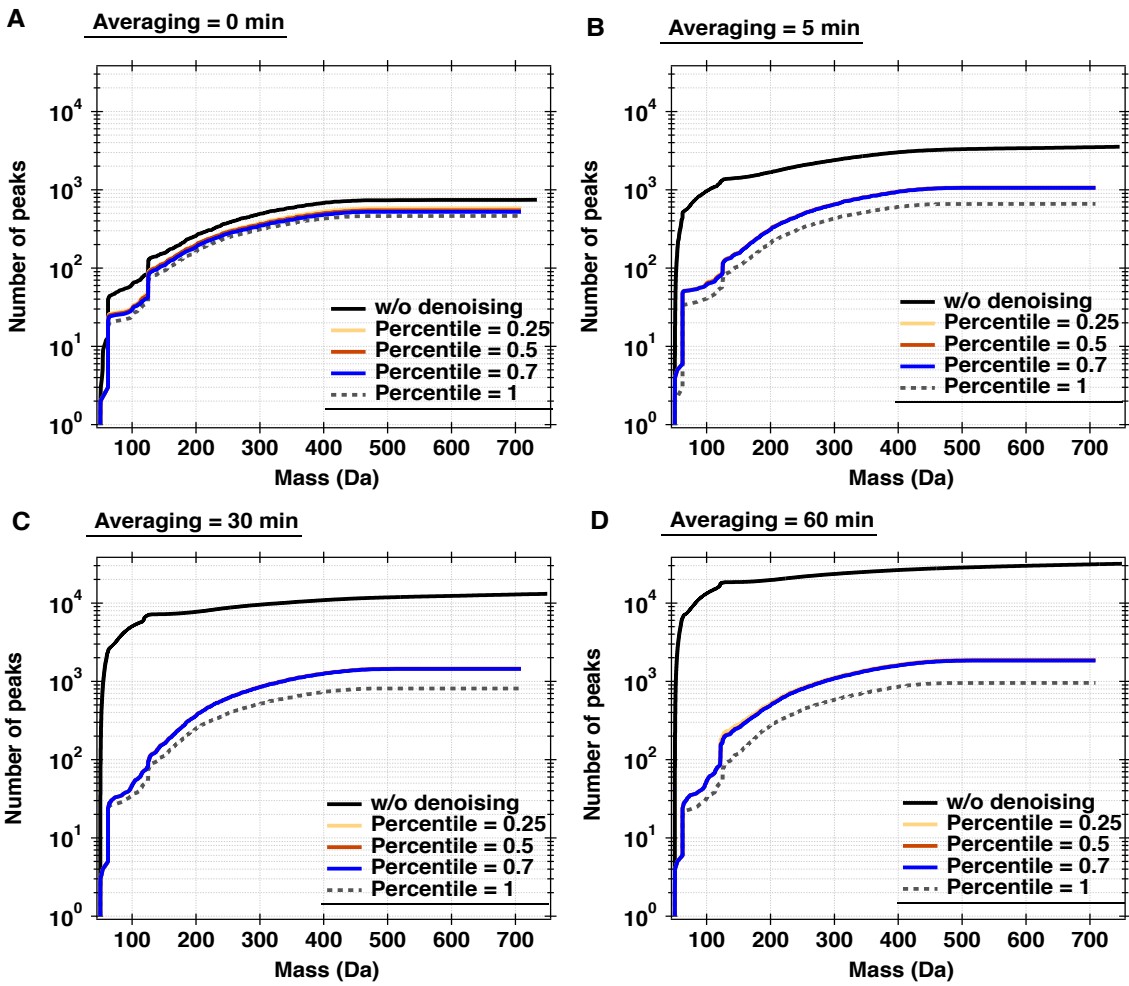

**Figure 3.** Number of peaks automatically detected in ambient air by Orbitool with and without noise reduction and for different averaging time. When the percentile = 100%, all the peaks in the mass defect range of [0.5, 0.8] was removed.

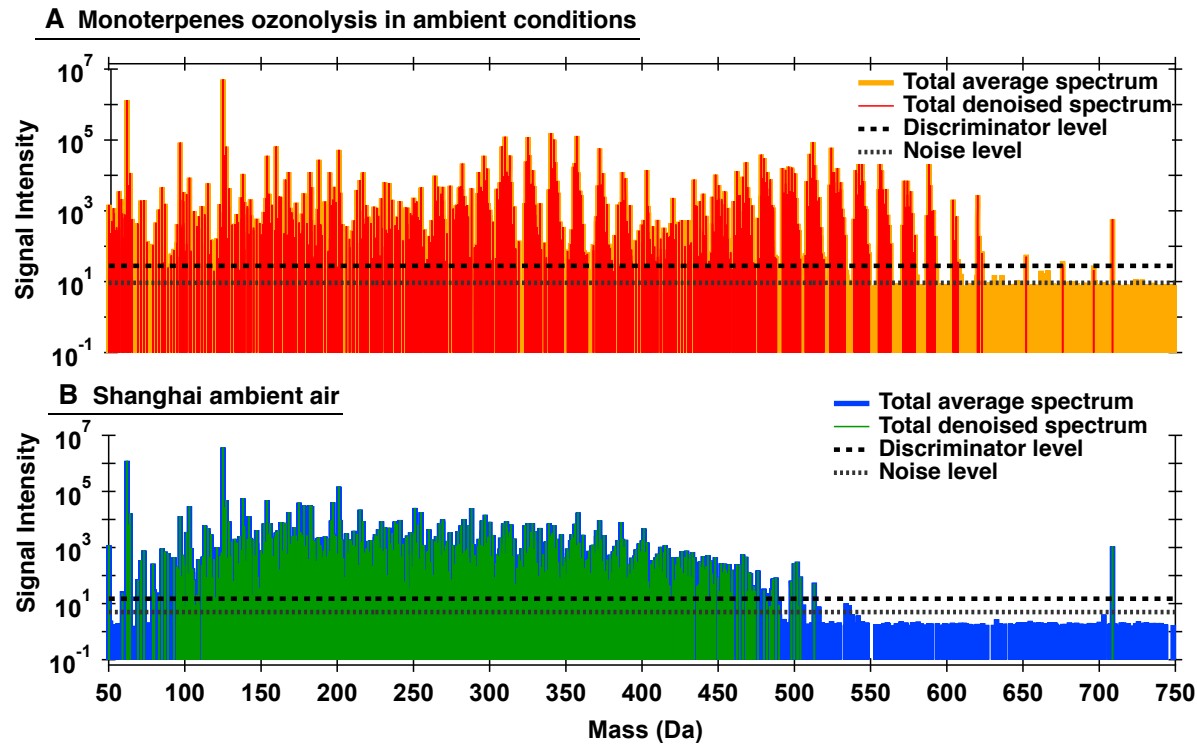

**Figure 4:** The noise and discriminator level of mass spectra obtained from the ozonolysis of monoterpenes in **(A)** ambient indoor air conditions (via peeling an orange in front of the Orbitrap) and **(B)** ambient air measured in Shanghai (31/10/2019 between 14:00-14:30). The averaging span for raw data in **(A)** and **(B)** were 5 and 30 min, respectively. The 70th percentile was used to estimate the noise and discriminator level.

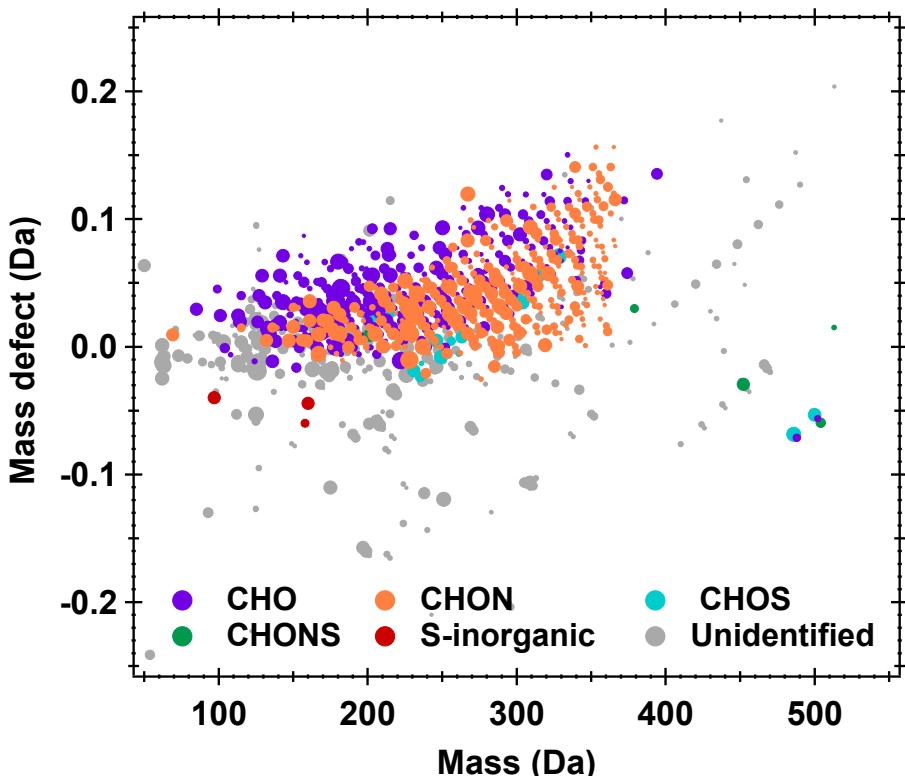

**Figure 5.** Mass defect of gaseous compounds observed using the CI-Orbitrap in Shanghai (31/10/2019 between 14:00-14:30). The mass defects larger than 0.5 are subtracted by 1.0 so that the domain of mass defect in this figure is [-0.5, 0.5].