# Peer review of "Orbitool: A software tool for analyzing online Orbitrap mass spectrometry data"

_Atmospheric Measurement Techniques, 2020_

## Referee Comment (RC1) · Anonymous Referee #3 · 14 Sep 2020

General comments:

The authors present the application of a novel software solution (open source) for analyzing Thermo Fisher raw mass spectrometry files, which are not designed for online monitoring. This is of great value for the atmospheric science community, which increasingly applies Orbitrap technology for atmospheric applications. The paper certainly fits into the scope of AMT.

However, the paper requires quite some language editing and needs to explain some approaches more in detail. I am surprised that the major benefit of a HR-MS is not shown: the capability to resolve isobaric species.

Furthermore, the authors should comment on the data acquisition of the Orbitrap,

which pre-filters data and should only record signals that are not attributed to (electronic) noise. Otherwise, the reader might be confused by the extensive effort of distinguishing noise from signal.

The term "identification [of compounds]" is used in this manuscript quite often, but it rather describes the attribution of a molecular formula to a measured mass. An accurate mass measurement (incl. isotopic pattern) can only help in determining the sum formula, not the identification of compounds! For the correct use of "molecular identification" see Noziere et al., Chem. Rev., 2015. Therefore, the abstract needs rewriting, since the terms "identification" and "separation" are discussed in a wrong context.

Since Orbitool is GUI-based software, some words about installation instruction on the website can be helpful for researchers which are not familiar with Python, but still are willing to use the software.

Overall, the manuscript fits into AMT very well, the work is important for the community, but the paper needs additional information, major rewriting and some corrections, as partly listed under the following comments.

Specific comments:

p.2 l.8: Do you really mean "noise"? IUPAC definition of noise: "The random fluctuations occurring in a signal that are inherent in the combination of instrument and method." Maybe your analysis just discards noise with a more strict filter than the XCalibur acquisition software (which to my knowledge already applies a noise-filter during data acquisition)?

p.2 l.9: The presented work does not show ozonolysis of monoterpenes, as atmospheric scientists would think of. You tested ozonolysis of orange peel emissions. These emissions contain monoterpenes, but not exclusively.

p.2 l.13: "Identification of unknown species" is not in line with the molecular identification defined by Noziere et al. − it is rather "sum formula attribution" than "identification".

p.2 l.15: In this case, do not use the term "separate" in order to avoid misunderstanding with chromatographic separation techniques.

p.5 l.10: TofTools also requires the background data between all nominal masses, which are not recorded by XCalibur.

p.6 l.7: Before describing how data are handled with Orbitool, it should be described how data are recorded (E.g. experimental setup, ion source settings, data acquisition settings (scan rate, pre-averaging, use of a lock-mass, centroid mode, profile mode?), etc.).

p.7, l.9: I do not understand what the numbers of the mass defect range ([0.5, 0.8]) intend to express. What is the center of your mass defect and what is the isolation width. This does not become clear.

p.8, l.22: The concept of the lock mass is that the mass accuracy is stable for long time-series, making additional mass calibrations obsolete.

p.9, l.9-16: For ion signals which are > 1e6 molecules cm-3, the isotopic pattern can be used to verify / falsify sum formulas by calculating an isotopic pattern matching score. Is this feature possible with Orbitool?

p.11, l.16: Additionally to the mass defect plot, other visualizations might be also informative, such as the aromaticity index or the Kendrick mass defect.

p.11, l.22: Again, I do not understand the mass defect range of [0.5, 0.8] as a filter for determining the noise level. Does this rather broad range of 0.5 amu requires only one main ion signal within this range? My experience, is that in such a large mass range on can usually find more than five-to-ten different (baseline-separated) ion signals. I think the text needs a more detailed explanation.

p.12, l.10-12: ". . . number of peaks after noise reduction with the 50th percentile is

insensitive to the averaging time." –> I cannot see this in the data: After 5 min averaging time the number of peaks converges to ∼1000, after 30 min to ∼1300, and after 60 min to ∼2000. Hence, your reasoning appears questionable.

Figure 4: Can you also demonstrate the Orbitrap technology really has an advantage over ToF-MS by resolving several gas-phase signals on one nominal mass? My experience is that with NO3-CIMS (using ToF) of monoterpene ozonolysis peaks are already well fitted, and show little evidence for isobaric interference. A zoom in on the x-scale of both experiments would be worth to show.

Technical corrections:

Consider to minimize the use of "greatly" (used in p.2 l. 8, l. 13)

p.2 l.3: "wildly-used"? I think you rather intended to say "widely-used"

p.2 l.6: maintaining –> improving?

p.2 l.14: consider: . . . ambient gas-phase measurements in urban Shanghai.

p.3 l. 2: produce . . . into the atmosphere? Needs rephrasing. E.g. Biogenic and anthropogenic sources emit a wide variety of VOCs into the atmosphere.

p.4. l. 22-23: What do you mean with ". . .overcome the interference of noise and accumulate signals."?

p.8. l.20: It is m/z which is determined, not the mass.

Figure 5: blue and purple are very hard to distinguish from each other.
* * *

---

## Referee Comment (RC2) · Anonymous Referee #2 · 15 Sep 2020

Summary and recommendation:

In this study, Cai et al. describe the development of a new software tool for the analysis of Orbitrap data, focusing in particular on data treatment procedures that are common in atmospheric sciences. The authors demonstrate the usefulness of the developed procedures and explain how the algorithms process the data. Moreover, they show examples of laboratory and field data that were processed using the new software. The software is planned as an open source tool and freely available on the internet after user registration.

In my opinion, the authors did a very good job and created an impressive tool for Orbitrap data analysis, which follows routines that are familiar to the CIMS community. I agree with the authors that Orbitrap mass spectrometers will play a more and more

important role in atmospheric sciences in the future. Therefore, a software tool tailored to the needs of online Orbitrap MS data is highly desirable and will certainly help CI-TOFMS users to transition to CI-Orbitrap MS.

The manuscript is well-written and explains in a reader-friendly manner all modules of the software. Therefore, I have only some minor comments, which should be addressed before final publication in AMT.

Minor comments:

1) P6L17f: I cannot follow the authors explanation on the averaging weight for the spectra. Could you explain in more detail why the scan number is more important than the duration of a single scan?

2) P7L5: How do the authors define background here? Is it only electronic background? As far as I know there is already a built-in background subtraction of signals by the acquisition software from the manufacturer.

3) P7L9f: For readers less familiar with the common procedures in the CIMS community, it might be helpful to explain shortly why the noise estimation is conducted in the mass defect range of 0.5-0.8 Da.

4) P8L13f: The description of the peak width normalization should be revised. Currently, it remains unclear why the peaks are normalized to 200 Da after dividing them by the square root of m.

5) P9L13: As far as I know, the default setting of XCalibur is to remove lock masses and reference signals from the mass spectra. Could the authors check again?

6) P10L4ff: It remains unclear whether isotopic patterns of candidate formulae are considered in the formula assignment procedure. If they are not considered so far, I would suggest to include this in future versions of the software. Even if the isotopic signals might deviate at low concentrations from the expected intensity, it would still be reasonable to check for the presence of such signals. Could the authors comment on

this?

7) P12L19: Over which time span were the spectra averaged that are shown in Fig. 4?

8) P13L10f: Could the authors provide a list of observed signals and assigned molecular formulas in the supporting material? Moreover, do the authors suggest the presence of gas-phase organosulfates and nitrooxy organosulfates which are among the CHOS and CHONS compounds?

9) P14L15ff: Several authors are missing in the list of "Author Contributions" (i.e., C.G., M.E., C.H., and P.Y.)

Technical comments:

1) Abstract / L2: replace "wildly-used" with "widely-used"

2) P3L1: replace "produce" with "emit" (or a similar verb)

3) P6L15: replace "within each time bin" with "for each time bin"

4) P6L23: better use "ion polarity" instead of "charge polarity"

5) P7L1: replace "positively charged spectra" with "spectra from positive mode" (and accordingly for "negatively charged spectra")

6) P8L22: word missing after "this"

7) P10L6: better use "ion polarity" instead of "charge polarity"

8) P10L20 and P11L11: replace "exported into file" by "exported into other file formats"

9) P11L8: replace "These" by "The"

10) P12L14: remove "50th"

11) P13L1: replace "then" by "the"

12) Figure 3: It is difficult to distinguish the colors of the different percentiles

13) Figure 4: typos in the legend ("denoized" should read as "denoised")

---

## Short Comment (SC1) · 15 Sep 2020

**Myriam Guillevic**

myriam.guillevic@empa.ch

Received and published: 15 September 2020

We would like to make a brief technical comment regarding naming of the released code/version.

in Cai et al., the authors write that the newly developed software is Open Source. Technically, this means that the source code (.py files in Python, as written by the authors), should be made available, so that other people can check each step of the process and read the algorithms. However, as also mentioned by the reviewers, so far what is available is an executable version to install, after registration. Also on the project webpage <https://orbitrap.catalyse.cnrs.fr/>, it is clearly stated that " an executable Orbitool is available". We downloaded the zip file and we found indeed only the executable, with

Python bytecode (.pyd and .dll files).

It would be good if the authors clarify the description of what is exactly made available to registered users.

If the authors wish to make only an executable available, to avoid other scientists to check or re-use the source code, this is called a 'Freeware' version, i.e. users have no access to the source code but the executable is available for free. If you do not make the source code available, you cannot call it 'Open Source', please use 'Freeware' instead, so that users don't get confused.

Or, if the authors plan to release the source code along with the executable in the future, please mention this clearly (e.g., where the source code will be hosted).

---

## Referee Comment (RC3) · Anonymous Referee #4 · 30 Sep 2020

General comment:

The paper presents Orbitool, a novel online software tool for analyzing Orbitrap mass spectrometry data. The paper is within the scope of AMT but is often too superficial in the description of the working principles of the software and there is no real validation of the output. Some specific comments follow.

Specific comments:

p.4, l.1: the mass resolving power of the best TOF analyzers is now more than 50,000. https://www.sciencedirect.com/science/article/pii/S0165993619302018

p.4, l.7-13: this paragraph on source comparison does not belong to this software development article

p.5, l.20: I do not understand what is meant by "long-term atmospheric measurements". Is it related to the averaging feature?

p.6, l.10: even if the orbitrap was recording all the data points, this would not generate an equally spaced grid since the size of the bins (i.e. resolution) is variable with the mass

p.6, l.12: the term "proper averaging" needs to be clarified: what are the different methods to make the average?

p.6, l.15: "time bin" - this concept seems important afterwards but it is not defined

p.6, l.17-18: why making weighted averages?

p.6, l.19-20: why would the mathematical operation of average involve the resolution of the Orbitrap?

p.6, l.19-23: this should be at the beginning of the paragraph; first describe the procedure before going into the details of how it's done and not the other way around

p.7, l.8: the mathematical effect of the average itself causes the noise level to go down (division of a constant level by a higher number of spectra), not the instrumental fact of acquiring more time

p.7, l.8-9: I am guessing that the mass defect range of [0.5, 0.8] Da was chosen because no signal peaks are expected with such a mass defect in the mass range studied?

p.7, l.13: what is "a certain percentile"? I had to wait until p.12, l.16-17 to find the explanation. It should be in the description section, not in the the discussion. See also p.12, l.5-6

p.7, l.17: here, I think it is necessary to develop why the averaged spectrum would have different noise thresholds because according to me, even if the individual spectra have different noise thresholds, the resulting spectrum will in any case have a single

noise level from a statistical point of view. Actually, I don't understand the difference in the final result between option b and option a

p.8, l.1-3: I understand that the orbitrap doesn't generate signals at 0 for the masses it doesn't detect. However, I don't see the link between removing noise, i.e. removing data, and the fact that the orbitrap doesn't populate empty masses. It seems obvious that removing data (noise) reduces the size of the file and thus the number of points to be processed afterwards

p.8, l.9-10: my personal experience is that peaks are not really Gaussian, and I don't seem to see any discussion on why the Gaussian model works in your case. It seems necessary to develop what you say in the caption in the body of the text

p.10, l.9-10: this statement needs a reference or at least more explanations

p.12, l.22: Th here and Da in figures. The same unit should be used throughout the paper

p.13, l.14-15: I'm having a hard time with this sentence (and this paragraph, actually). It's a software article. A software needs to be validated with a systematic validation protocol with other software/publications and the quantification of the performance. It cannot just be like "oh, that's cool, we are seeing things we've already seen and things we don't know anything about"

p.13, l.21: is it Orbitool or rather the fundamental difference between ToF and Orbitrap data? Because nowhere is there any mention of a data storage strategy that is not based on the fact that Orbitrap data is generated in the right way

p.14, l.1: I am not convinced. See my previous comment above

p.14, l5-7: This sentence does not belong to this article

Figure 5: The mass defect as presented in this figure does not match with the definition of the mass defect in p.7, l.9-10: depending on your definition of the nominal mass, it

should be either in the range of [0, 1] or of [-0.5, 0.5]

---

## Author Comment (AC1) · 4 Nov 2020

**Responses to Comments on Manuscript amt-2020-267**

**(Orbitool: A software tool for analyzing online Orbitrap mass spectrometry data)**

We thank Dr. Myriam Guillevic for her concerns on the public availability of Orbitool. As clarified below, Orbitool is an open-source code. In the following paragraphs, the comments are shown as sans-serif dark red texts and our responses are shown as serif black texts.

**Reviewer #1, Dr. Myriam Guillevic**

We would like to make a brief technical comment regarding naming of the released code/version.

In Cai et al., the authors write that the newly developed software is Open Source. Technically, this means that the source code (.py files in Python, as written by the authors), should be made available, so that other people can check each step of the process and read the algorithms. However, as also mentioned by the reviewers, so far what is available is an executable version to install, after registration. Also on the project webpage <https://orbitrap.catalyse.cnrs.fr/>, it is clearly stated that " an executable Orbitool is available". We downloaded the zip file and we found indeed only the executable, with Python bytecode (.pyd and .dll files).

It would be good if the authors clarify the description of what is exactly made available to registered users.

If the authors wish to make only an executable available, to avoid other scientists to check or re-use the source code, this is called a 'Freeware' version, i.e. users have no access to the source code but the executable is available for free. If you do not make the source code available, you cannot call it 'Open Source', please use 'Freeware' instead, so that users don't get confused.

Or, if the authors plan to release the source code along with the executable in the future, please mention this clearly (e.g., where the source code will be hosted).

**Response**: Both the source codes (.py files) and the executable software (.exe file) are now publically available on the website.

We have added the following text on the website:

"Researchers who are not familiar with Python can directly run the Orbitool.exe file without a Python programming language on their computer. An executable can be downloaded using the following link…

Source codes (.py files) are publically available in the following files…"

In addition to providing open-source codes, we aim to facilitate researchers to use Orbitool in a more flexible way. Researchers who are experienced with Python programming can modify Orbitool and add new features by themselves. They can also use the Orbitool functions without following the standing data processing procedure or relying on the graphical user interface. We are currently refactoring the codes and adding brief comments so that it will take less effort to understand the codes. Classes and functions will be defined in a more general way to facilitate customized applications.

---

## Author Comment (AC2) · 4 Nov 2020

**Responses to Comments on Manuscript amt-2020-267**
**(Orbitool: A software tool for analyzing online Orbitrap mass spectrometry data)**

We thank the reviewer for the comments that help to improve this manuscript. The manuscript was revised according to these comments. We added more details in the revised manuscript. The reviewer's comments are addressed in the following paragraphs. The comments are shown as sans-serif dark red texts and our responses are shown as serif black texts. Changes are highlighted in the revised manuscript and shown as "quoted underlined texts" in the responses. Reference are given at the end of the responses.

**Reviewer #4**

**General comment:**

The paper presents Orbitool, a novel online software tool for analyzing Orbitrap mass spectrometry data. The paper is within the scope of AMT but is often too superficial in the description of the working principles of the software and there is no real validation of the output. Some specific comments follow.

**Response**: We added more details on the working principles in the revised manuscript.

To validate the software, we have shown that the Orbitool is able to read and average raw spectra, fit a certain distribution to the profile of single peaks, conduct mass calibration, distinguish peaks and their corresponding abundances via peak fitting, assign molecular formulae to fitted peaks, etc. We agree with the reviewer's argument that they are not real validation; however, a real validation needs known true values as the benchmark, which is impossible for all the validations. Besides, the data processing methods used in Orbitool are mostly based on basic algebra and/or existing methods for the analysis with CI-ToF data. Therefore, we do not think the results presented in this manuscript lacks validation.

Specific comments:

p.4, l.1: the mass resolving power of the best TOF analyzers is now more than 50,000.

https://www.sciencedirect.com/science/article/pii/S0165993619302018

**Response**: We agree with the reviewer that TOF with greater mass resolving exists. However, to our knowledge the greatest mass resolving power for an online mass spectrometer is 15 000. TOF analyzers with a mass resolution of 50 000 are not suited to the needs of some applications in atmospheric researches (or have not been tested yet). For instance, to achieve such a mass resolving power, multi reflection technique is needed, which is often associated with poor ion transmission and sensibility. We revised this sentence as: "The mass resolving power of a TOF analyzer typically ranges from hundreds to less than 50 000 and the mass resolving power of online TOF-MS, used in atmospheric measurements, is only up to 15 000."

p.4, l.7-13: this paragraph on source comparison does not belong to this software development article

**Response**: This paragraph is not a comparison of different ionization sources. Instead, it introduces the potential applications of Orbitrap mass spectrometer in atmospheric researches. The contribution of Orbitool to the research

community is affected by the broadness of its potential applications. As a result, it is important to first introduce these potential applications.

p.5, l.20: I do not understand what is meant by "long-term atmospheric measurements". Is it related to the averaging feature?

**Response**: This phrase was replaced by "long-term field or laboratory campaigns". Data averaging, noise reduction, mass calibration, and exporting time series are closed related to the demands of long-term field campaigns. "Long-term" indicates the amount of data to be analyzed, the importance of calculating and exporting time series over a period (e.g., a week or a month), and the importance of mass accuracy. "Atmospheric measurements" indicates low signal intensities and hence the importance of data averaging and noise reduction. These demands have been discussed in the Introduction section.

p.6, l.10: even if the orbitrap was recording all the data points, this would not generate an equally spaced grid since the size of the bins (i.e. resolution) is variable with the mass

**Response**: Thanks. This phrase was revised as "a dense grid-based spectrum".

p.6, l.12: the term "proper averaging" needs to be clarified: what are the different methods to make the average?

**Response**: We removed "proper" and added "The data recorded in the same file is averaged using an averaging function in the Thermo RawFileReader library". We compared this averaging algorithm with RawFileReader, an averaging method from a third-party python package, and averaging methods coded by us. RawFileReader is used for its accuracy and low computational expense. However, we prefer not to present these details in the manuscript to keep it concise.

p.6, l.15: "time bin" - this concept seems important afterwards but it is not defined

**Response**: We replaced it with "time interval for averaging".

p.6, l.17-18: why making weighted averages?

**Response**: This sentence was rewritten as "the average is calculated with respect to scan number". The original sentence wanted to emphasize that the average is calculated for scan number, yet the weight of each scan is equal.

p.6, l.19-20: why would the mathematical operation of average involve the resolution of the Orbitrap?

**Response**: Because two data points needed to be determined as the same one or two different ones before averaging. For instance, there are two data point, $(x_1, y_1)$ and $(x_2, y_2)$, measured at two different time, where x is the mass and y is the signal intensity. Before averaging, the software needs to decide whether $x_1 \approx x_2$. If the y values are stored with predetermined fixed array of x, the criterion for the approximate equality is determined by the round-off error of the used data type and it can be practically written as $x_1 = x_2$. However, the Orbitrap does store data in this way. Even for two adjacent scans, the values of $x_1$ and $x_2$ corresponding to the same data point are slightly different and this difference may increase with time. Hence, a criterion is needed to determine whether $|x_1-x_2|$ is no larger than this criterion. If so, the averaged x is the weight average of $x_1$ and $x_2$; otherwise, $(x_1, y_1)$ and $(x_2, y_2)$ are saved as two different data points. This criterion is determined by grid of the x array (although Orbitrap data is not save as a continuous array), and the density of the grid is determined by the mass resolution.

 this should be at the beginning of the paragraph; first describe the procedure before going into the details of how it's done and not the other way around

**Response**: We agree with the reviewer that the general procedure should be described before introducing the details. However, l.19-23 in the original manuscript are on how data points are identified as the same or different, when the averaging functions are triggered, and advanced features of data averaging. They are the details rather than general procedures. The general features, i.e., configuring data files for averaging, starting and end time, and temporal resolution for averaged data, are introduced at the beginning of this paragraph.

p.7, l.8: the mathematical effect of the average itself causes the noise level to go down (division of a constant level by a higher number of spectra), not the instrumental fact of acquiring more time

**Response**: We did not say "acquiring more time" or anything similar.

p.7, l.8-9: I am guessing that the mass defect range of [0.5, 0.8] Da was chosen because no signal peaks are expected with such a mass defect in the mass range studied?

**Response**: Following this sentence, we wrote "Such a mass defect range is chosen because most of the observed compounds in atmospheric measurements are located outside of this mass range." Since some signals may locate in this range, we select only a proportion, not all, of peaks in this range to estimate the noise threshold.

p.7, l.13: what is "a certain percentile"? I had to wait until p.12, l.16-17 to find the explanation. It should be in the description section, not in the the discussion. See alsop.12, l.5-6

**Response**: We added "This certain percentile can be customized and it is by default 70%, i.e., the 70% low peaks within this mass defect range are taken as noise peaks and used to estimate the noise level."

p.7, l.17: here, I think it is necessary to develop why the averaged spectrum would have different noise thresholds because according to me, even if the individual spectra have different noise thresholds, the resulting spectrum will in any case have a single noise level from a statistical point of view. Actually, I don't understand the difference in the final result between option b and option a

**Response**: We revised "different amounts of noise" as "different amounts of noise peaks" to avoid confusion. The three strategies are on how to use the noise level instead of how to obtain it.

The difference between options a and b is whether the signals is subtracted by the noise level, as clarified in the manuscript. This subtraction mainly affects the intensity of peaks with intensities slightly above the discriminator level. For instance, when the noise level is 1 and the signal peak height is 4, there is a 25% difference for this peak between options a and b (4 and 3, respectively). When the noise peaks locates sparsely, the measured height of 4 is most likely to be contributed by the signals and option a is recommended. In contrast, when the noise peaks is dense after averaging for a long period, the measured peak height of 4 is most likely to be a combination of the signal peak and the noise. In this case, option b is recommended.

p.8, l.1-3: I understand that the Orbitrap doesn't generate signals at 0 for the masses it doesn't detect. However, I don't see the link between removing noise, i.e. removing data, and the fact that the orbitrap doesn't populate empty masses.

It seems obvious that removing data (noise) reduces the size of the file and thus the number of points to be processed afterwards

**Response**: We do not understand this comment because we did not mention Orbitrap here. Removing noise refers to the procedure to remove the noise peaks after reading from file and averaging. Practically, it reduces the length of the array to store the averaged spectrum but not the size of the raw data file. The raw data file is readable only for Orbitool.

p.8, l.9-10: my personal experience is that peaks are not really Gaussian, and I don't seem to see any discussion on why the Gaussian model works in your case. It seems necessary to develop what you say in the caption in the body of the text

**Response**: Figure 2A shows that the normal distribution is consistent with the measured peak profiles. In the main text, we wrote "As shown in Fig. 2A, the normalized peaks can be well characterized using a normal distribution."

We have not encountered a failure of the normal distribution for our datasets. Other fitting functions can be readily added to the Orbitool, in case there is such a demand in the future.

p.10, l.9-10: this statement needs a reference or at least more explanations

**Response**: We have revised the sentence as follow: "If needed, the nitrogen rule can be used to help constraining the peak identification (Pellegrin, 1983)."

p.12, l.22: Th here and Da in figures. The same unit should be used throughout the paper

**Response**: We revised "Th" as "Da" throughout the manuscript.

p.13, l.14-15: I'm having a hard time with this sentence (and this paragraph, actually). It's a software article. A software needs to be validated with a systematic validation protocol with other software/publications and the quantification of the performance. It cannot just be like "oh, that's cool, we are seeing things we've already seen and things we don't know anything about"

**Response**: We disagree with the reviewer that a software article should exclude application examples. To its contrary, an application example provides a straight forwarding understand of the capability of this software. Besides, the beginning phrase "To illustrate the capabilities of Orbitool…" indicates that this paragraph is not on the validation of the software.

As aforementioned, the key scientific features of Orbitool is tested and reported in this manuscript. The validation of algorithms is either reported if they are not only based on simple algebra. True values are naturally absent for a real validation. The usual validation routines mainly focus on the technical part of software development, e.g., bug and error management, and hence we prefer not to include them in this manuscript. Finally, it is worth mentioning that bugs and error managements are mentioned in the readme file that is available when downloading the software.

p.13, l.21: is it Orbitool or rather the fundamental difference between ToF and Orbitrap data? Because nowhere is there any mention of a data storage strategy that is not based on the fact that Orbitrap data is generated in the right way

**Response**: The data structure of a spectrum can be easily changed upon reading from the raw data file. For instance, if the number of data points is relatively low after filling the blanks of an Orbitrap spectrum, ToFTools is probably able to

analyze Orbitrap data after some modifications. However, due to the much greater mass resolving power of the Orbitrap, such a grid-based data structure will cause an ultra-high computational expense. Note that a different data structure corresponds to different algorithms, classes, functions, and variable, i.e., a different software.

In short, the data structures of Orbitrap raw data and the spectrum class in Orbitool both result from the high resolving power of Orbitrap and there is no causal relationship between the data structures of Orbitrap raw data and Orbitool.

p.14, l.1: I am not convinced. See my previous comment above

**Response**: We removed "successfully". We agree with the reviewer that this part is an application example instead of a validation.

p.14, l5-7: This sentence does not belong to this article

**Response**: We removed this sentence.

Figure 5: The mass defect as presented in this figure does not match with the definition of the mass defect in p.7, l.9-10: depending on your definition of the nominal mass should be either in the range of [0, 1] or of [-0.5, 0.5]

**Response**: We added "The mass defects larger than 0.5 are subtracted by 1.0 so that the domain of mass defect in this figure is [-0.5, 0.5]" in the caption of Fig. 5. The range of vertical axis shown in Fig.5 is [-0.25, 0.25] because no molecular formulae are identified outside this range. This is consistent with the algorithm to estimate the noise level using the mass defect range of [-0.5, -0.2] (i.e., [0.5, 0.8]).

---

## Author Comment (AC3) · 4 Nov 2020

**Responses to Comments on Manuscript amt-2020-267**
**(Orbitool: A software tool for analyzing online Orbitrap mass spectrometry data)**

We thank the reviewer for the constructive comments that help to improve this manuscript. The manuscript was revised correspondingly. The reviewer's comments are addressed in the following paragraphs. The comments are shown as sans-serif dark red texts and our responses are shown as serif black texts. Changes are highlighted in the revised manuscript and shown as "quoted underlined texts" in the responses. References are given at the end of the responses.

**Reviewer #2**

Summary and recommendation:

In this study, Cai et al. describe the development of a new software tool for the analysis of Orbitrap data, focusing in particular on data treatment procedures that are common in atmospheric sciences. The authors demonstrate the usefulness of the developed procedures and explain how the algorithms process the data. Moreover, they show examples of laboratory and field data that were processed using the new software. The software is planned as an open source tool and freely available on the internet after user registration.

In my opinion, the authors did a very good job and created an impressive tool for Orbitrap data analysis, which follows routines that are familiar to the CIMS community. I agree with the authors that Orbitrap mass spectrometers will play a more and more important role in atmospheric sciences in the future. Therefore, a software tool tailored to the needs of online Orbitrap MS data is highly desirable and will certainly help CITOFMS users to transition to CI-Orbitrap MS.

The manuscript is well-written and explains in a reader-friendly manner all modules of the software. Therefore, I have only some minor comments, which should be addressed before final publication in AMT.

**Response**: We thank the reviewer for these positive comments. To facilitate atmospheric researches with Orbitrap mass spectrometers with various needs, Orbitool will be updated continuously and a forum will be shortly open to collect user's demands.

Minor comments:

1) P6L17f: I cannot follow the authors explanation on the averaging weight for the spectra. Could you explain in more detail why the scan number is more important than the duration of a single scan?

**Response**: This sentence was revised as: "The data recorded in the same file is averaged using an averaging function in the Thermo RawFileReader library and the average is calculated with respect to scan number."

The Thermo RawFileReader library calculates the average spectrum with respect to scan number. Xcalibur$^{TM}$ uses the same averaging function. The duration of a single scan does not vary much or the total ion current is relatively stable. However, an implicit assumption is that the parameters cannot be changed during the time interval of an averaged spectrum. We added the following sentence in the revised manuscript: "Hence, it is assumed that the parameters determining the duration of every single scan (e.g., injection time and the number of micro scans) are kept constant during each time interval."

2) P7L5: How do the authors define background here? Is it only electronic background?

As far as I know there is already a built-in background subtraction of signals by the acquisition software from the manufacturer.

**Response**: Thanks for this comment. We added the following text to better explain how the background is treated. "When converting electronic signals into a spectrum via the Fourier transform, the instrument removes signals below the noise threshold defined by the Orbitrap. Signals exceeding this noise threshold are converted into peaks recorded in the data file. Since this procedure does not remove all the noise signals, some noise peaks are also recorded in the data file. In this study, "noise" refers to these noise peaks." To avoid confusion, we removed "background" when referring to noise.

3) P7L9f: For readers less familiar with the common procedures in the CIMS community, it might be helpful to explain shortly why the noise estimation is conducted in the mass defect range of 0.5-0.8 Da.

**Response**: There was a short explanation below this sentence: "Such a mass defect range is chosen because most of the observed compounds in atmospheric measurements are located outside of this mass range."

4) P8L13f: The description of the peak width normalization should be revised. Currently, it remains unclear why the peaks are normalized to 200 Da after dividing them by the square root of m.

**Response**: We revised this sentence as follow: "According to this relationship, the measured widths of these selected peaks are normalized to widths at 200 Da by dividing them by $\sqrt{m/200 \text{ Da}}$."

We also added the following information: "200 Da is chosen as the reference value because the instrumental resolution of the Orbitrap is usually reported at this mass."

5) P9L13: As far as I know, the default setting of XCalibur is to remove lock masses and reference signals from the mass spectra. Could the authors check again?

**Response**: We agree with the reviewer and rewrite this sentence to avoid confusion.

The revised sentence is: "Instead of excluding the lock masses from the spectra (i.e., the default setting in Xcalibur), Orbitool treats the lock masses as normal signals since the abundance of the reagent ion is usually a critical parameter to characterize the performance of a CI-mass spectrometer."

6) P10L4ff: It remains unclear whether isotopic patterns of candidate formulae are considered in the formula assignment procedure. If they are not considered so far, I would suggest to include this in future versions of the software. Even if the isotopic signals might deviate at low concentrations from the expected intensity, it would still be reasonable to check for the presence of such signals. Could the authors comment on this?

**Response**: Orbitool has an isotope list for less abundant isotopes and the isotope pattern can be used for peak determination.

To clarify this, we added a new paragraph: "Orbitool also supports the identification of isotopes. The users can adjust the possible isotopes in an isotope list and all the isotopes in this list will be considered. Using the default algorithm for peak determination, the chemical formulae containing a less abundant isotope can be determined only when its corresponding formula containing the abundant isotope is found in the spectrum. In addition, this default algorithm also checks whether the abundance ratio of these two species is consistent with their natural abundances within the uncertainty range. To

facilitate studies such as isotope labeling experiments, Orbitool also provides an algorithm that does not restrict isotope abundances during peak determination. However, it is important to mention that the Orbitrap provides non-linear responses when the concentrations of the analytes are very low (i.e., $< 1 \times 10^6$ molecules cm$^{-3}$, at a 10-minute integration time). As a result, the calculated isotope abundance may deviate from its true value (Riva et al., 2020)."

7) P12L19: Over which time span were the spectra averaged that are shown in Fig. 4?

**Response**: We added "measured in Shanghai (31/10/2019 between 14:00-14:30)" to the caption of Figure 4.

8) P13L10f: Could the authors provide a list of observed signals and assigned molecular formulas in the supporting material? Moreover, do the authors suggest the presence of gas-phase organosulfates and nitrooxy organosulfates which are among the CHOS and CHONS compounds?

**Response**: We do not claim that we observed organosulfates and nitrooxy organosulfates. What the molecular assignment is showing is the presence of sulfur-containing compounds (CHOS/CHONS) in the gas phase. Such compounds might be clusters of carboxylic acid, sulfuric acid and amine (e.g., $(C_xH_yO_z)(H_2SO_4)NO_3^-$ or $(C_xH_yO_z)(H_2SO_4)(C_mH_nN)NO_3^-$) as such clusters are expected to exist in polluted environments with high concentrations of precursors (Lin et al., 2019). Earlier studies have also reported the presence of organosulfur compounds in the gas phase (e.g., Ehn et al., 2010). However more work is needed to determine the sources as well as the chemical processes forming such species, but such investigation is beyond the scope of this study. We are currently working on separate studies to characterize the composition and sources of the gaseous species observed during this field campaign.

9) P14L15ff: Several authors are missing in the list of "Author Contributions" (i.e., C.G.,M.E., C.H., and P.Y.)

**Response**: Thanks. We added their contributions in the revised manuscript.

Technical comments:

1) Abstract / L2: replace "wildly-used" with "widely-used"

2) P3L1: replace "produce" with "emit" (or a similar verb)

3) P6L15: replace "within each time bin" with "for each time bin"

4) P6L23: better use "ion polarity" instead of "charge polarity"

5) P7L1: replace "positively charged spectra" with "spectra from positive mode" (and accordingly for "negatively charged spectra")

6) P8L22: word missing after "this"

7) P10L6: better use "ion polarity" instead of "charge polarity"

8) P10L20 and P11L11: replace "exported into file" by "exported into other file formats"

9) P11L8: replace "These" by "The"

10) P12L14: remove "50th"

11) P13L1: replace "then" by "the"

12) Figure 3: It is difficult to distinguish the colors of the different percentiles

13) Figure 4: typos in the legend ("denoized" should read as "denoised")

**Response**: We thank the reviewer for these suggestions. The manuscript was corrected accordingly.

---

## Author Comment (AC4) · 4 Nov 2020

**Responses to Comments on Manuscript amt-2020-267**
**(Orbitool: A software tool for analyzing online Orbitrap mass spectrometry data)**

We thank the reviewer for the constructive comments that help to improve this manuscript. The manuscript was revised according to these comments. We added more details in the revised manuscript and polished the language. The reviewer's comments are addressed in the following paragraphs. The comments are shown as sans-serif dark red texts and our responses are shown as serif black texts. Changes are highlighted in the revised manuscript and shown as "quoted underlined texts" in the responses. Line numbers, figures, and equations quoted in the responses correspond the revised manuscript. Reference are given at the end of the responses.

**Reviewer #3**

The authors present the application of a novel software solution (open source) for analyzing Thermo Fisher raw mass spectrometry files, which are not designed for online monitoring. This is of great value for the atmospheric science community, which increasingly applies Orbitrap technology for atmospheric applications. The paper certainly fits into the scope of AMT.

However, the paper requires quite some language editing and needs to explain some approaches more in detail. I am surprised that the major benefit of a HR-MS is not shown: the capability to resolve isobaric species.

**Response**: Thanks. We added more details in the revised manuscript. The terms used in this manuscript was revised according to the reviewer's comments to improve the accurateness and readability of this manuscript.

We do not put the high-resolution peak signals retrieved using Orbitool in the manuscript because the high resolution is an advantage of the Orbitrap rather than Orbitool and similar figures have been reported in our previous studies (Riva et al., 2019 and 2020; Lee et al., 2020).

Furthermore, the authors should comment on the data acquisition of the Orbitrap, which pre-filters data and should only record signals that are not attributed to (electronic) noise. Otherwise, the reader might be confused by the extensive effort of distinguishing noise from signal.

**Response**: Thanks. In the revised manuscript, we added the following text: "When converting electronic signals into a spectrum via the Fourier transform, the instrument removes signals below the noise threshold defined by the Orbitrap. However, there are still signals exceeding this noise threshold and they are converted into peaks recorded in the data file. In this study, "noise" refers to the noise peaks recorded in the data file."

The term "identification [of compounds]" is used in this manuscript quite often, but it rather describes the attribution of a molecular formula to a measured mass. An accurate mass measurement (incl. isotopic pattern) can only help in determining the sum formula, not the identification of compounds! For the correct use of "molecular identification" see Noziere et al., Chem. Rev., 2015. Therefore, the abstract needs rewriting, since the terms "identification" and "separation" are discussed in a wrong context.

**Response**: We thank the reviewer for this clarification. In the revised manuscript, we replaced "identification of chemical compounds" with "assignment of molecular formula". "Separate" was replaced by "distinguish" when it refers to two chemical formulae.

Since Orbitool is GUI-based software, some words about installation instruction on the website can be helpful for researchers which are not familiar with Python, but still are willing to use the software.

**Response**: We have added some description: "The users can either run Main.py via Python or Orbitool.exe without Python. The required Python environment (optional) for Orbitool is described in detail on the website" to ***Software availability***.

A brief instruction on the Python version and required packages has been added to the latest version of Orbitool.zip file, which can be found on the website.

Researchers who are not familiar with Python can directly run the Orbitool.exe file without a Python programming language on their computer.

Overall, the manuscript fits into AMT very well, the work is important for the community, but the paper needs additional information, major rewriting and some corrections, as partly listed under the following comments.

**Specific comments:**

p.2 l.8: Do you really mean "noise"? IUPAC definition of noise: "The random fluctuations occurring in a signal that are inherent in the combination of instrument and method." Maybe your analysis just discards noise with a more strict filter than the XCalibur acquisition software (which to my knowledge already applies a noise-filter during data acquisition)?

**Response**: We agree with the reviewer that "noise" in this manuscript does not follow the usual definition. It refers to the signal peaks generated by remained noises after the pre-filtering before the Fourier transform. To clarify it, we added the following text: "When converting electronic signals into a spectrum via the Fourier transform, Orbitrap applies a noise threshold to remove noises. However, there are still noises exceeding this noise threshold and they are converted into peaks recorded in the data file. In this study, "noise" refers to the noise peaks recorded in the data file. It is difficult to separate the signal peaks for some compounds with low concentrations from noise peaks."

p.2 l.9: The presented work does not show ozonolysis of monoterpenes, as atmospheric scientists would think of. You tested ozonolysis of orange peel emissions. These emissions contain monoterpenes, but not exclusively.

**Response**: We used the VOCs emitted from an orange peels to have a simple and user-friendly method to mass calibrate the Orbitrap (i.e., using the ambient $O_3$ to oxidize VOC, including monoterpenes emitted from an orange). We do not claim that an orange only emits monoterpenes, but the formation of HOMs has been mainly observed from the oxidation of monoterpenes (at least with high yields). As shown in Figure R1, the mass spectrum obtained from such mass calibration test is very similar to a laboratory-controlled monoterpene ozonolysis experiment, as reported in our initial study or by other groups (e.g., Jokinen et al., 2015). As a result, such simple approach can be very useful and easy to carry out without the need of dedicated experimental setup.

[Figure]

Figure R1, a mass spectrum from orange peeling experiment.

p.2 l.13: "Identification of unknown species" is not in line with the molecular identification defined by Noziere et al. – it is rather "sum formula attribution" than "identification".

**Response**: We agree and replace "identification" with "assignment of molecular formula". Molecular formula, by its definition, does not distinguish isomers.

p.2 l.15: In this case, do not use the term "separate" in order to avoid misunderstanding with chromatographic separation techniques.

**Response**: This phase was revised as "assign hundreds of molecular formulae".

p.5 l.10: TofTools also requires the background data between all nominal masses, which are not recorded by XCalibur.

**Response**: We added "In addition, TofTools determines the noise level using the equally spaced data within a certain mass defect range, whereas such information is not recorded in the Orbitrap data" to this paragraph.

p.6 l.7: Before describing how data are handled with Orbitool, it should be described how data are recorded (E.g. experimental setup, ion source settings, data acquisition settings (scan rate, pre-averaging, use of a lock-mass, centroid mode, profile mode?), etc.).

**Response**: We thank the reviewer for this suggestion. The current version of Orbitool is developed for data recorded in the profile mode. Other parameters do not affect the analysis procedure with Orbitool, although they affect the analysis results and may influence the difficulty to analyze a spectrum data (for example, using a lock-mass may reduce the difficulties in attributing molecular formula to a peak). In the revised manuscript, we added the following text: "Most tuning parameters of Orbitrap, e.g., injection time of each micro scan and the number of micro scans for each single scan, do not affect the analysis with Orbitool. However, it should be specially clarified that Orbitool is compatible with the profile mode but not the centroid mode of Orbitrap, because the attribution of chemical formulas to measured signal peaks is based on fitting peak distributions to the profile data."

p.7, l.9: I do not understand what the numbers of the mass defect range ([0.5, 0.8]) intend to express. What is the center of your mass defect and what is the isolation width. This does not become clear.

**Response**: If the mass defect corresponding to the peak mass of a fitted peak is no smaller than 0.5 and no larger than 0.8, it is selected in this step of data analysis. [0.5, 0.8] is a standard expression of a closed interval. The sentence was revised as follow: "Orbitool first takes all the detected peaks whose peak positions are located in the mass defect range of [0.5, 0.8] Da."

p.8, l.22: The concept of the lock mass is that the mass accuracy is stable for long time-series, making additional mass calibrations obsolete.

**Response**: We are not certain that the additional mass calibration is always redundant. For example, when using the reagent ions and its dimers of a CI-Orbitrap as lock masses, a majority of the measured species locates at larger masses. As a result, the mass accuracy of these measured species with large masses is improved by extrapolation rather than interpolation. An additional mass calibration with more species may contribute to the mass accuracy.

As a software tool to facilitate customized data analysis, Orbitool can skip the mass calibration procedure. This feature can be used when additional mass calibration is unnecessary.

The relevant sentence was revised as "The Orbitrap is able to maintain its mass accuracy (i.e., < 2 ppm) for long-time series..." and we added "The user can also skip the mass calibration."

p.9, l.9-16: For ion signals which are > 1e6 molecules $cm^{-3}$, the isotopic pattern can be used to verify / falsify sum formulas by calculating an isotopic pattern matching score. Is this feature possible with Orbitool?

**Response**: Orbitool supports this feature. For each assigned molecular formula, its theoretical relative abundance is displayed in the widget for peak fitting and molecular formula assignment. If this molecular formula contains a less abundant isotope and its corresponding molecular formula with the more abundant isotope exists in the spectrum, the measured relative abundance is also displayed. The value of the measured relative abundance is by default used as a criterion during the peak assignment. The measured relative abundance cannot exceed of certain ratio of the theoretical relative abundance, otherwise Orbitool will not assign this molecular formula. Meanwhile, Orbitool provides a separate peak assignment algorithm that does not use the isotope abundance to filer possible molecular formulae.

In the revised manuscript, we added a paragraph to discuss the details of this feature related to isotope abundance: "Orbitool also supports the identification of isotopes. The users can adjust the possible isotopes in an isotope list and all the isotopes in this list will be considered. Using the default algorithm for peak determination, the chemical formula containing a less abundant isotope can be determined only when its corresponding formula containing the abundant isotope is found in the spectrum. In addition, this default algorithm also checks whether the abundance ratio of these two species is consistent with their natural abundances within the uncertainty range. To facilitate studies such as isotope labeling experiments, Orbitool also provides an algorithm that does not restrict isotope abundances during peak assignment."

p.11, l.16: Additionally to the mass defect plot, other visualizations might be also informative, such as the aromaticity index or the Kendrick mass defect.

**Response**: Thanks. We will add these features in future versions of the Orbitool.

p.11, l.22: Again, I do not understand the mass defect range of [0.5, 0.8] as a filter for determining the noise level. Does this rather broad range of 0.5 amu requires only one main ion signal within this range? My experience, is that in such a large mass range on can usually find more than five-to-ten different (baseline-separated) ion signals. I think the text needs a more detailed explanation.

**Response**: The reviewer is correct that multiple peaks may be found in this broad mass range. This range is expressed in a closed interval of [0.5, 0.8] and hence the width is 0.3 amu for each unit mass. The corresponding sentence in the manuscript is "Orbitool first takes all the detected peaks whose peak positions are located in the mass defect range of [0.5, 0.8] Da", which should have no ambiguity. The aim of this step is to the potential noise peaks instead of a single noise peak. The noise level is determined by statistics on these selected peaks, as elaborated in section 2.2.

p.12, l.10-12: ". . . number of peaks after noise reduction with the 50th percentile is insensitive to the averaging time." –> I cannot see this in the data: After 5 min averaging time the number of peaks converges to _1000, after 30 min to _1300, and after 60 min to _2000. Hence, your reasoning appears questionable.

**Response**: We revised this sentence as: "This is consistent with the observed trend of the total number of peaks after noise reduction in Fig. 3, in which the total number of peaks after noise reduction with the 50th percentile increases slightly with an increasing averaging time." This observed relationship between the number of peaks and averaging time is consistent with the previous sentence "Considering that some species may not be detected in a single scan (i.e., present at very low concentration), the total number of peaks in the averaged spectrum should grow with the increasing averaging time when the period is considerably short, and then it should converge to a certain value when the averaging period is sufficiently long."

Figure 4: Can you also demonstrate the Orbitrap technology really has an advantage over ToF-MS by resolving several gas-phase signals on one nominal mass? My experience is that with $NO_3$-CIMS (using ToF) of monoterpene ozonolysis peaks are already well fitted, and show little evidence for isobaric interference. A zoom in on the x-scale of both experiments would be worth to show.

**Response**: As discussed earlier in this review, the manuscript focuses on the software part (i.e., Orbitool) but not on the advantages of the hardware (i.e., Orbitrap). In addition, such figures have already been reported in all of our previous studies (Riva et al., 2019 and 2020; Lee et al., 2020).

**Technical corrections:**
Consider to minimize the use of "greatly" (used in p.2 l. 8, l. 13)
p.2 l.3: "wildly-used"? I think you rather intended to say "widely-used"
p.2 l.6: maintaining –> improving?
p.2 l.14: consider: . . . ambient gas-phase measurements in urban Shanghai.
p.3 l. 2: produce . . . into the atmosphere? Needs rephrasing. E.g. Biogenic and anthropogenic sources emit a wide variety of VOCs into the atmosphere.

**Response**: Thanks. We revised this manuscript according to the above comments.

p.4. l. 22-23: What do you mean with ". . .overcome the interference of noise and accumulate signals."?

**Response**: This sentence was revised as: "The measured spectra must be averaged over a long period to decrease the noise level so that these low signals can be unambiguously identified among noises."

p.8. l.20: It is m/z which is determined, not the mass.

**Response**: We prefer to avoid using m/z (i.e., mass/charge) as mass and charge are both dimensional quantities. Because they have different dimensions, mass/charge is a dimensional quantity. As a result, we prefer to keep a meaningful annotation and decide to use mass (Da) which is also consistent with our previous published recently in AMT (Riva et al., 2019).

Figure 5: blue and purple are very hard to distinguish from each other.

**Response**: Thanks. We changed the colors of markers.

**References**

Jokinen, T., Berndt, T., Makkonen, R., Kerminen, V-M., Junninen, H., Paasonen, P., Stratmann, F., Herrmann, H., Guenther, A. B., Worsnop, D. R., Kulmala, M., Ehn, M. K., & Sipila, M.: Production of extremely low volatile organic compounds from biogenic emissions: Measured yields and atmospheric implications, Proc. Natl. Acad. Sci. U. S. A., 112(23), 7123-7128. https://doi.org/10.1073/pnas.1423977112, 2015

Lee, C. P., Riva, M., Wang, D., Tomaz, S., Li, D., Perrier, S., Slowik, J. G., Bourgain, F., Schmale, J., Prevot, A. S. H., Baltensperger, U., George, C. and El Haddad, I.: Online Aerosol Chemical Characterization by Extractive Electrospray Ionization–Ultrahigh-Resolution Mass Spectrometry (EESI-Orbitrap), Environ. Sci. Technol., 54(7), 3871–3880, doi:10.1021/acs.est.9b07090, 2020.

Riva, M., Ehn, M., Li, D., Tomaz, S., Bourgain, F., Perrier, S. and George, C.: CI-Orbitrap: An Analytical Instrument To Study Atmospheric Reactive Organic Species, Anal. Chem., 91(15), 9419–9423, doi:10.1021/acs.analchem.9b02093, 2019.

Riva, M.; Rantala, P.; Krechmer, J. E.; Peräkylä, O.; Zhang, Y.; Heikkinen, L.; Garmash, O.; Yan, C.; Kulmala, M.; Worsnop, D. Evaluating the Performance of Five Different Chemical Ionization Techniques for Detecting Gaseous Oxygenated Organic Species. Atmos. Meas. Tech., 12, 2403–2421, doi.org/10.5194/amt-12-2403-2019, 2019.

Riva, M., Brüggemann, M., Li, D., Perrier, S., George, C., Herrmann, H. and Berndt, T.: The capability of CI-Orbitrap for gas-phase analysis in atmospheric chemistry: A comparison with the CI-APi-TOF technique, Anal. Chem., acs.analchem.0c00111, doi:10.1021/acs.analchem.0c00111, 2020.